# Protective Application of Chinese Herbal Compounds and Formulae in Intestinal Inflammation in Humans and Animals

**DOI:** 10.3390/molecules28196811

**Published:** 2023-09-26

**Authors:** Yang Yang, Gang Xiao, Pi Cheng, Jianguo Zeng, Yisong Liu

**Affiliations:** 1College of Veterinary Medicine, Hunan Agricultural University, Changsha 410125, China; yangyang0616@foxmail.com (Y.Y.); xiaogang2020@stu.hunau.edu.cn (G.X.); picheng@hunau.edu.cn (P.C.); 2Hunan Key Laboratory, Chinese Veterinary Medicine, Changsha 410125, China

**Keywords:** intestinal inflammation, cellular signaling pathway, Chinese herbal compounds, Chinese herbal formulae, gut health

## Abstract

Intestinal inflammation is a chronic gastrointestinal disorder with uncertain pathophysiology and causation that has significantly impacted both the physical and mental health of both people and animals. An increasing body of research has demonstrated the critical role of cellular signaling pathways in initiating and managing intestinal inflammation. This review focuses on the interactions of three cellular signaling pathways (TLR4/NF-κB, PI3K-AKT, MAPKs) with immunity and gut microbiota to explain the possible pathogenesis of intestinal inflammation. Traditional medicinal drugs frequently have drawbacks and negative side effects. This paper also summarizes the pharmacological mechanism and application of Chinese herbal compounds (Berberine, Sanguinarine, Astragalus polysaccharide, Curcumin, and Cannabinoids) and formulae (Wumei Wan, Gegen-Qinlian decoction, Banxia xiexin decoction) against intestinal inflammation. We show that the herbal compounds and formulae may influence the interactions among cell signaling pathways, immune function, and gut microbiota in humans and animals, exerting their immunomodulatory capacity and anti-inflammatory and antimicrobial effects. This demonstrates their strong potential to improve gut inflammation. We aim to promote herbal medicine and apply it to multispecies animals to achieve better health.

## 1. Introduction

IBD (inflammatory bowel disease) is a chronic, relapsing gastrointestinal ailment that affects people worldwide today. It comes in two primary forms: Crohn’s disease and ulcerative colitis (UC). Kaplan carried out a review, reporting that the highest prevalence values were in Europe and Norway, as well as the rapidly rising incidences in Asia, Africa, and South America since 1990 [1,2]. Furthermore, due to the large population in conjunction with expanding urbanization and Westernization, the number of cases of IBD in newly industrialized countries such as India and China might overtake the Western world at some point. Thus, IBD is a global disease [3]. Patients with IBD have seriously affected their quality of life, suffering from gastrointestinal symptoms such as diarrhea and bloody stool, and they also have extraintestinal complications such as arthropathies and mucocutaneous. The consensus is that multiple factors contribute to IBD, including the innate and adaptive immune response, genetic susceptibility, abnormal gut microbiota response, epigenetic change, and environmental factors [4]. In recent years, biological therapies targeting cytokines have demonstrated significant clinical efficacy against IBD, and they could interact with inflammatory signaling pathways involved in IBD to achieve inhibition of the inflammatory response. For instance, TNF could trigger a cascade of intracellular responses, which would subsequently cause the activation of inflammatory signals such as JAK/STAT and NF-κB [5]. Compared to traditional IBD drugs (5-aminosalicylic and azathioprine), monoclonal antibodies (mAbs) revolutionized the IBD treatment and had impacts on lowering recurrence and maintaining remission. In Asian countries, more and more clinical applications of mAbs have been reported, and prevailing mAbs are divided into several categories, including anti-TNF drugs, integrin blockers, and anti-IL-12/IL-23 therapies [6]. Therefore, biological agents targeting inflammatory signaling pathways are part of therapeutic approaches [7,8]. However, it is a reality that the uptake of biological agents has gradually increased. Many IBD patients switch between different biologic agents frequently, and clinical discontinuation of biologic agents is uncommon. Unfortunately, biologic therapy has a high annual cost per patient. In addition, immunomodulators and biological medicines that weaken the immune system over an extended period of time increase the risk of lymphoma and opportunistic infections [3]. This is why there is an urgent need for biological alternatives to alleviate IBD symptoms and the economic situation of patients.

In the poultry industry, gut health is the predominant state for flock performance and morbidity. However, a persistent gut inflammatory response will result in disturbed digestion, constant oxidative stress, and impaired immunological competence, which affects the farm’s economic returns [9]. Gut inflammation is triggered by multiple environmental factors in poultry commercial production, including animal density, intestinal pathogens, high-energy diets, and poor-quality feed ingredients [10,11]. A malabsorption model in broiler chickens was induced by rye as the only energy source to identify intestinal health biomarkers in the serum. These findings showed that the body weight (BW) and daily weight gain (DWG) of the rye-fed chickens were significantly reduced, and citrulline and IFN-γ levels in serum as well as IgA levels in cloacal tissues showed significant differences, suggesting that they may be the potential biomarkers for intestinal inflammation [12]. After the bacterial cocktail consisting of *Escherichia coli*, *Enterococcus faecalis*, *Lactobacillus salivarius*, *Lactobacillus crispatus*, *Clostridium perfringens (netB)*, and *Ruminococcus gnavus*, as well as a coccidial suspension consisting of *Eimeria acervulina* and *Eimeria maxima*, the chicken’s intestines were attacked. The main manifestations were lower BW and DWG, as well as severe intestinal lesions [13]. For chronic intestinal inflammation, the current data demonstrated that the lower small intestine was affected later than the upper, and the challenged duodenum and jejunum were attacked earlier than the ileum, which evolved in a spatial and temporal pattern. Michael further pointed out that Lipocalin-2 (LCN-2) could be a fecal biomarker for intestinal inflammation, and more importantly, calprotectin was a marker with great potential as a novel biomarker in the serum and feces of chickens [14]. Chronic intestinal inflammation in broilers is largely caused by adaptive immune cells, particularly T lymphocyte infiltration, which is linked to higher levels of IL-16 and CCL20 production [15]. Heat stress also influences poultry performance and immune function via aggravating intestinal inflammation through the TLR4/NF-κB signaling pathway, making chickens more susceptible to *Escherichia coli* O157:H7 [16]. After the LPS challenge, the response of the duodenal was driven by intestine inflammation, and Paneth cells were impaired at the early stage but quickly recovered. It is well known that Paneth cells are essential for the development of intestinal stem cells (ISC), the base of the rapid renewal and repair of the intestinal mucosa. Hence, Paneth cells modulate ISC activity to improve the mucosal barrier during the initiation of intestinal inflammation in chickens [17].

Traditional Chinese medicine (TCM), which is an ancient medicine with a complete medical system that cures illnesses through evidence-based treatment, is one of the most advanced subspecialties of complementary and alternative medicine. TCM has contributed to Chinese medicine, characterized by extensive resources and low cost. It is boiled into a tonic to treat diseases based on different medicinal properties and portions of the drug, so its active ingredients are not easily lost and its side effects are relatively mild. Chinese medicine has different theories and approaches to treating intestinal inflammation. According to TCM, IBD falls under the category of “diarrhea” or “dysentery,” which includes conditions like damp-heat in the large intestine, cold-heat syndrome, spleen deficiency and dampness, spleen-kidney yang deficit, as well as liver depression and spleen deficiency. In recent years, there has been an increase in the number of studies to explain the mechanism of herbal medicine in IBD treatment. The methods offered by the TCM range encompass a vast number of herbal concoctions with previously unknown compositions and mechanisms of action. With the advancement of technology nowadays, a large number of natural chemicals present in TCM are attracting attention due to their distinctive advantages of having few adverse effects, sustained efficacy, and a broad range of targets [18]. Interestingly, TCM can be applied flexibly by adjusting the dosage based on the individual and their symptoms [19]. Herbal compounds and Chinese herbal formulae can be used to alleviate IBD symptoms through anti-inflammatory, antioxidant, anti-bacterial, and immune-boosting effects [20]. This paper intends to summarize the inflammatory signaling pathways related to intestinal inflammation and the effects of Chinese herbal compounds and formulae against intestinal inflammation, aiming to strengthen the understanding of intestinal inflammation and open the mind to the application of Chinese medicines to the intestinal health of humans and animals.

## 2. Cellular Signaling Pathways in Intestinal Inflammation

The protective application of Chinese herbal compounds and formulae on intestinal inflammation is exerted mainly through six main aspects, including the improvement of IBD pathology symptoms, regulation of the intestinal microbiota, protection of the intestinal barrier function, reduction of inflammatory cytokines, improvement of oxidative stress, regulation of immunity, and regulation of cellular signaling pathways. Numerous studies have proven that different signaling pathways, the immune system, and the balance of the gut microbiota have important roles in the development of the progression of IBD, and Chinese herbal compounds and formulae can exert ameliorative and therapeutic effects on intestinal inflammation through the interaction of multiple signaling pathways, immunity, and the gut microbiota [21,22].

### 2.1. TLR4/NF-κB

Toll-like receptor 4 (TLR4) belongs to the TLR family of receptors that are activated by lipopolysaccharide (LPS) and expressed in non-immune cells such as intestinal epithelial cells and endothelial cells. Intestinal inflammation exhibits dysbiosis of the intestinal flora and immune system. LPS is a major component of the outer membrane of Gram-negative bacteria, and TLR4 responds to the rising expression of LPS and triggers the pro-inflammatory response of the organism with pathogen clearance. In this time frame, the specific mechanism is the binding of LPS to the LBP protein, its translocation from the bacterial outer membrane, and aggregation around the receptor, followed by CD14-mediated formation of the LPS-LBP-CD14 complex and binding to TLR4/MD-2 [23]. The TLR receptor family recruits a series of adaptors with TIR domains, such as MyD88, TRIF, TIRAP/MAL, and TRAM. The two types of TLR signal routes are MyD88-dependent and TRIF-dependent, depending on the adaptor applications. After activation of the MyD88-dependent signaling pathway, through a series of signal transmission and ubiquitination modifications [24], TAKI is stimulated to bind to the IKK complex, resulting in phosphorylation of the IKK complex, activation of IKKβ and inhibition of IκBα, translocation of NF-κB nuclear, and induction of pro-inflammatory cytokines (IL and TNF-α) expression [25] (Figure 1).

#### 2.1.1. Related to Immunity

TLRs are thought to be among the most ancient pathogen recognition systems, and TLR4 is known for its unique ability to distinguish PAMPs from various types of infections. TLR4 stimulation triggers a series of adaptive immune reactions with the goal of providing the body with the greatest level of protection while causing the least amount of tissue damage [29]. Monocytes are the circulating precursors and dendritic cells, performing tasks like antigen presentation. Monocytes produce a number of inflammatory cytokines (e.g., IL-6, IL-1β, TNF-α) triggered by TLR4 activation in inflammatory settings [30]. Surprisingly, Sacramento et al. demonstrated that the TLR4 signaling pathway prevented the development of the Th1 immune response through an interferon regulatory factor 1 (IRF1)- and interferon-alpha/beta receptor (IFNAR)-dependent mechanism. This is an immunomodulatory mechanism that controls chronic inflammatory processes [31]. In general, LPS is a potent TLR4/MD-2 receptor complex agonist, resulting in the transcription of genes linked to immune responses that are pro-inflammatory. However, the weak agonistic LPS is responsible for the observed healing effects under intestinal inflammation [32]. While it continues to activate TLR4 and NF-κB, it actively attempts to restore intestinal immunological homeostasis in mice situated in intestinal inflammation, providing a fundamental anti-inflammatory intracellular transcription program without exceeding a pro-inflammatory threshold [33]. Remarkably, lysine acetylation is assumed to be essential for the transcription of inflammatory factors via modulation of histone acetyltransferases (HATs) and histone deacetyl transferases (HDACs), and TLR4 acetylation is associated with innate immune regulation. According to certain research, lysine acetylation may affect LPS-mediated gene expression profiles in macrophages and dendritic cells [34]. In LPS-induced inflammatory responses, HDACs inhibit the TLR4/NF-κB pathway by reducing NF-κB acetylation to limit the severity of the inflammation. Moreover, different HDACs have diverse effects on NF-κB-mediated inflammatory responses, such as trichostatin A, which could reduce inflammation by repressing a few NF-κB target genes in macrophages but aggressively activate NF-κB-mediated inflammatory responses in microglia cells [35]. Immune responses mediated by TLR4/NF-κB signaling are intricate overall.

#### 2.1.2. Related to Gut Microbiota

TLRs are important immunological receptors for bacteria, which they use to regulate intestinal inflammation. According to B. Otto et al.’s research, TLR4-deficient IL-10-/- mice that were infected with *Campylobacter jejuni* showed a less impaired intestinal barrier, less live commensal gut bacteria translocation, and less pronounced intestinal pro-inflammatory immune responses, as indicated by fewer B lymphocytes in the colonic mucosa [36]. M. M. Heimesaat et al. proved that TLR4 is essential for pro-inflammatory immune responses elicited by *Arcobacter butzleri* in the intestinal tract, which manifested as reduced amounts of ileal T lymphocytes, Tregs, and B cells, as well as less pronounced IFN-γ secretion when TLR4 was deleted [37]. A further study revealed that the absence of TLR4 could enhance commensal enterobacteria and enterococci in the intestinal tract while decreasing fecal obligate anaerobic *Clostridium*/*Eubacterium spp*. These adjustments, in turn, made it easier for *Pseudomonas aeruginosa* to colonize and made it more difficult to get rid of it. It was found that by 14 days, the spleen’s inflammatory and immunological levels were the exact opposite of what occurred in the colon [38]. Acute pancreatitis and gut injury were evident in intestinal epithelia TLR4 deletion animals, which might be explained by a reduction in Paneth cells and a disruption in the intestinal flora, particularly the depleted *Lactobacillus* [39]. Taken together, TLR4/NF-κB is a potential molecular mechanism in the emergence of intestinal inflammation based on gut microbiota [40].

### 2.2. PI3K-AKT

Phosphatidylinositol kinases (PI3Ks) are crucial for cell survival, cytoskeleton remodeling, metabolic regulation, and other activities. PI3K signaling is mainly based on the study of class I PI3Ks, a family of intracellular signaling enzymes that phosphorylate phosphatidylinositol (PtdIns), with the end product PtdIns (3,4,5) P3 targeting AKT [26]. AKT is a protein kinase B, a serine/threonine kinase. After phosphorylation of the Thr308 and Ser473 positions, AKT on the cell membrane is completely activated and transferred to the cytoplasm or nucleus, which in turn phosphorylates numerous target proteins to carry out biological functions [27] (Figure 1).

#### 2.2.1. Related to Immunity

PI3K signaling is not a simple switch for cellular activation but rather an intricate network of interactions that must maintain cellular immune homeostasis. The PI3K family of lipid kinases regulates diverse processes involved in lymphocyte activation and development. In T cells, both class IA and class IB PI3K are the dominant subgroups that contribute to development and immunological activity. Class IA PI3K is critical for B cells in determining development and antigen responsiveness [41]. Emerging evidence supports the hypothesis that PI3K plays a crucial role in activating the lymphocytes, and one sign of this is a shift in the expression of chemokine receptors and homing receptors, which makes it easier for activated cells to be transported to peripheral organs. In CD8 T cells, PI3K activation promotes both shedding and transcriptional shut-off of the lymph node homing receptor, L-selectin (CD62L), through a route involving Erk, whereas transcriptional regulation is mTORC1-dependent [42]. Lymphocytes use PI3K-Erk-mTOR to match trafficking. Yuan et al. verified that the deletion of CCR3 reduced eosinophilic inflammation and the Th2 immune response by inhibiting the PI3K-AKT pathway in allergic rhinitis mice, and targeted knockout of CCR3 inhibited the expression of the PI3K-AKT signaling pathway in eosinophils and the proliferation, migration, and degranulation of eosinophils in vitro experiments [43]. PI3K-AKT signaling fine-tunes the inflammatory response in macrophages by balancing the induction of signaling pathways that positively and negatively regulate inflammation, and blunted PI3K-AKT activation helps to explain the phenotypic dysfunction associated with macrophages by regulating the Cdc42/Rac effector kinase, including hyper-inflammation, defective bacterial clearance, and defective migration [44,45]. In lymphocytes, cytokine or chemokine signaling causes PI3K activation, resulting in AKT activation. Much of the studies on AKT in macrophages focuses on the phenotypes M1 and M2. According to the experiments with macrophages lacking either AKT1 or AKT2, it was proposed that whereas the ablation of AKT1 gives rise to M1, the ablation of AKT2 gives rise to M2 [46]. AKT may modulate the inflammatory state because M1 macrophages are proinflammatory while M2 exhibits anti-inflammatory actions [47].

#### 2.2.2. Related to Gut Microbiota

Class IA PI3Ks have three p110 catalytic subunits, of which p110δ is mainly enriched in leukocytes [48]. Notably, the occurrence of spontaneous colitis was demonstrated in PI3K p110δKD mice. Furthermore, the presence of the enteric microbiota was necessary for the development of colitis in p110δKD mice. Interestingly, compared with germ-free (GF) WT mice, GF PI3K p110δKD mice produced considerably less colonic IL-10. This illustrates how the PI3K-AKT signaling pathway and colitis may be significantly influenced by the gut microbiota [49]. Commensal bacteria modulate intestinal inflammation and maintain intestinal homeostasis. In vivo and in vitro investigations showed that microbiota-induced IL-10-secreting intestinal B cells improved chronic LP CD4+ T cell-mediated colitis by activating TLR2/MyD88/PI3K p110δ signaling [50]. Gobert, A.P. et al. reported that *Helicobacter pylori* signals in macrophages to activate the PI3K and MTOR pathways, promoting the phosphorylation of AKT1 on Ser473 and the production of cystathionine L-lyase (CTH), which improved bacterial survival in macrophages by generating cystathionine to dampen the inflammatory and antimicrobial response. This is a novel strategy where bacterial pathogens can evade the innate immune response [51]. There was a study showing that the elevation of the PI3K and ERK1/2 signaling pathways was observed in malignant bronchial epithelial cell lines exposed to *Streptococcus*, *Prevotella*, and *Veillonella*, and was proven not to be triggered by LPS. It was conceivable that these pathogenic bacteria’s other compounds boosted PI3K signaling [52]. According to Yu et al.’s hypothesis, TL1A hyperactivation of PI3K/AKT/MLCK2-dependent bacterial endocytosis may result in gut microbes moving to extraintestinal viscera, triggering IBD symptom flares [53]. Some findings have been documented by both in vitro and in vivo studies, which probed that some strains of probiotics (e.g., *Bifidobacterium bifidum*, *Lactobacillus reuteri*, and LGG) inhibited the PI3K-AKT pathway to improve oxidative stress, support the intestinal autophagy machinery system, and repair the intestinal epithelial barrier [54].

In conclusion, it is complicated that the mechanism of the PI3K-AKT pathway regulates intestinal inflammation.

### 2.3. MAPKs

Mitogen-activated protein kinases (MAPKs), the protein serine/threonine kinases, convert extracellular stimulus signals into a wide range of cellular responses by phosphorylating diverse downstream targets [55]. There are conventional and atypical MAPKs in mammals. Conventional MAPKs are the most extensively studied groups of mammalian MAPKs, comprising the extracellular signal-regulated kinases 1/2 (ERK1/2), ERK5, c-Jun amino (N)-terminal kinases 1/2/3 (JNK1/2/3), as well as p38 isoforms (α, β, γ, and δ) [28]. These stimuli include cytokines, growth factors, hormones activating Ser/Thr kinase receptors, pathogen-associated molecular patterns (PAMPs) and danger-associated molecular patterns (DAMPs) that recruit pattern recognition receptors (PRRs), and environmental stresses. As said, the mechanisms of MAPK activation play an unexpectedly important role in inflammation and innate immunity [56,57]. The majority of these are, along with the nuclear NF-κB pathway, recruited by stress and inflammatory stimuli. MAPK-activated protein kinases MK2/3 are essential for the LPS-induced expression of pro-inflammatory cytokines and specifically promote the stress-activated cellular p38 response, and MK2/3 knockout mice show resistance against septic and endotoxic drugs [58]. MK2 and MK3 are promising targets for anti-inflammatory intervention. This idea is supported by the research confirming that ATI-450, the MK2 activation inhibitor, alleviated pathology and markers of colitis [59] (Figure 1).

#### 2.3.1. Related to Immunity

MAPKs serve as critical regulators in the maturation of T-lymphocytes and clonal expansion of T- and B-lymphocytes through modulation of cytokine synthesis, cell proliferation, and survival [60]. It is considered that p38 MAPK is a plausible contender to contribute to the differentiation of effector function between Th1 and Th2 cells. P38 MAPK is regulated by different cytokines, such as TNF-α, IL-1β, and IL-12, but it has also been involved in regulating cytokine gene expression and protein synthesis (e.g., IL-6, TNF-α). Cytokine production is the main characteristic that distinguishes the effector function of Th1 and Th2 and is also the major factor that determines the differentiation of naïve CD4+ T cells into effector Th1 and Th2 cells [61]. Rincon, M. et al. found that p38 MAPK was rapidly activated in Th1 cells but not in Th2 cells. Similar to p38 MAPK, the JNK pathway is not active in Th2 cells. It seems that Th2 may have a mechanism to prevent activation of both the JKN and p38 MAPK pathways. Possible explanations for this include the possibility that an upstream regulator for both the p38 and JNK pathways is present in Th1 cells but not in Th2 cells [62]. Drosophila is a well-established model for the study of the innate immune system and conserved signaling pathways. Anan Ragab et al. demonstrated that active Ras/MAPK signaling is necessary for intrinsic suppression of immune deficiency (IMD) signaling in Drosophila cells and in all immunological organs in vivo, including the adult midget. In addition, it was proposed that Ras/MAPK signaling functions in the immune system limit the intensity of IMD signaling during infection and avoid fictitious immune activation in the absence of illness [63]. The innate immune system needs to be tightly controlled because overreacting might result in acute or chronic inflammation. As a result, it is essential to stop MAPK signaling, and dual-specificity phosphatases (DUSPs) play a role in MAPK inactivation [64]. In DUSP1-/- macrophages, the p38 MAPK and JNK pathways are dysregulated, which has detrimental effects. Early on, LPS-induced DUSP1-/- macrophages, splenocytes, or entire animals overexpressed TNF and IL-10. Additionally, upregulation of other pro-inflammatory mediators, such as COX-2, IL-6, IL-1, CCL2, and CCL3, was observed in DUSP1-/- cells or animals at the mRNA or protein levels [65]. The pathogenesis of inflammatory disorders of the intestine is driven by inappropriate activation of innate immune responses in intestinal epithelial cells. When the p38 MAPK pathway is properly activated, organisms are protected from pathogen infection but suffer when there are no pathogens present [66].

#### 2.3.2. Related to Gut Microbiota

The increase or suppression of ERK1/2 and p38 MAPKs during infection with *Salmonella enterica serovar Typhimurium*, *Yersinia* spp., *Listeria monocytogenes*, and *Mycobacterium* spp. provides evidence that the MAPK signaling cascade is involved in bacterial pathogenesis. Additionally, B. suis-induced ERK1/2 MAPK pathway controlled bacterial death to equip host cells with a NO-generating system, which improved the cell’s ability to defend itself against bacteria [67]. To achieve inadequate clearance of apoptotic cells, *Caenorhabditis elegans* (*C. elegans*) stimulates the PMK-1 p38 MAPK and MPK-1/ERK MAPK pathways, which can support anti-microbial defense. For instance, apoptotic cell clearance mutant worms are more resistant to the pathogenic strains of *Salmonella typhimurium* SL1344 and Pseudomonas aeruginosa PA14 [68]. Ortiz, A. et al. discovered that *C. elegans*’ innate immune system reduces bacterial abundances through the p38 MAPK pathway but has little effect on some colonizers and the makeup of microbiotas with higher richness. It’s likely that the organism’s p38 pathway-dependent innate defense is specifically bacteriostatic [69]. The activation of MAPK by *Lactobacillus GG* was shown to regulate the activity of certain signaling pathways in intestinal epithelial cells. The expression of Hsp72, which is typically promoted by *Lactobacillus GG*, was prevented by the use of p38 and JNK inhibitors, and the peptides produced by *Lactobacillus GG* stimulated MAPK signaling for gut epithelial cells to protect them from oxidative stress and maintain cytoskeletal integrity [70]. Living involves adapting to a constantly changing environment, and this includes changes to the gut flora, which might have an impact on memory and behavior. In order to control *C. elegans*’ avoidance of Pseudomonas aeruginosa in the intestine, Kiho Lee et al. showed that transcription factor hlh-30 and the p38 MAPK pathway control the transcriptional expression of ins-11. Ins-11 is essential for maintaining homeostasis and shielding nematodes from perilous conditions to increase survival rates [71].

## 3. Chinese Herbal Compounds for Intestinal Inflammation

### 3.1. Berberine

Berberine (BBR) is a bioactive alkaloid extracted from several popular medicinal plants, such as *Coptidis chinensis*, *Phellodendron chinense*, and *Mahonia beale*. Following oral administration, BBR may have an enterohepatic circulation and may be absorbed in both the gastrointestinal tracts of rats and humans [72,73]. The part of BBR that was absorbed into the body could be converted into diversified metabolites. Gut microbiota acting as an “organ” could convert BBR into dhBBR in the intestine, mediated through bacterial nitroreductase. Additionally, dhBBR would oxidize in the colon and turn back into BBR [74]. This shows the importance and particularity of the gut to BBR.

BBR could promote the recovery of IBD, and this specific manifestation was to reverse the DSS-induced body weight loss, colon shortening, DAI, and myeloperoxidase (MPO) activity. Moreover, BBR reduced DSS-upregulated proinflammatory cytokines in the colon, including TNF, IFN-γ, IL-17, IL-1β, IL-6, and so on [75,76,77] (Table 1). More strikingly, by cellular isolation of colon tissue, we found that BBR has been shown to increase colonic macrophage apoptosis while decreasing apoptosis [75]. Yan et al. pointed out that the anti-IBD effect of BBR plays a role via MAPK and NF-κB signaling pathways [75]. Jing et al. suggested that the protective effects of BBR on colitis are attributed to the Nrf2-mediated overexpression of P-gp as a potential for IBD therapy in DSS-induced rats and in si-Nrf2 Caco-2 [76]. Takahara et al. explored BBR for IBD from multiple perspectives, showing that BBR might have an impact on the immune system and regulate CD4+ T cells, especially LP CD4+ T cells isolated from colitis SCID mice in vitro. Further studies confirmed that BBR inhibited the TH1/TH17-related JAK/STAT pathway of LP CD4+ T cells and mediated IFN-γ and IL-17A production by LP CD4+ T cells through AMPK activation in vivo experiments with colitis SCID mice and in vitro experiments with LP CD4+ T cells [78]. Guo et al. indicated that BBR could partially alleviate the gut microbiota disorder of IBD in cats and defined Lactobacillus as the most dominant species in BBR groups, which had a negative correlation with the DSS group. In addition, BBR reduces the colonic inflammatory response by inhibiting the TLR4/NF-κB pathway and activating MTORC and autophagy [79]. Li et al. revealed the effects of BBR on the gut microbiome in an in vitro model using metaproteomics and identified the enrichment of proteins from Verrucomicrobia, Proteobacteria, and Bacteroidetes phyla and the reduction of proteins from the Firmicutes phylum and its Clostridiales, which was correlated to the decline of the butyrate pathway and concentration [80].

Multiple highlights of crosstalk and communication between the immunological and neurological systems have been verified in a variety of research studies on IBD [81]. Li et al. reported that BBR modulated the production of neuropeptides from enteric glial cells (EGCs), which not only controlled the function of epithelial cells as barriers but also affected the interaction between intestinal epithelial cells (IECs) and immune cells. It was described as the BBR’s therapeutic mechanism, mediated by boosting EGC residency and modulating the crosstalk of enteric glial-immune-epithelial cells [77].

BBR supplementation enhanced the growth performance of yellow broiler, which may be positively correlated with the cecal microbiota composition, such as the abundance of Bacteroidetes at the phylum and genera levels, as well as Lactobacillus at the genera level [82]. Dehau et al. reported that BBR increased gut villus length and decreased crypt depth and CD3+ T-lymphocyte infiltration in chickens of various ages. However, the high dosage of BBR (2 g/kg) enriched the Enterobacteriaceae family and tended to reduce butyrate production in the cecum in vivo and in vitro. These results indicated that the high concentration of BBR supplementation improved chicken intestinal health, which was likely unrelated to changes in intestinal flora [83]. Studies showed that a high-carbohydrate diet (HCD) and a high-fat diet (HFD) could influence the intestine by causing inflammation, the immune system, oxidative stress, and flora composition [84,85]. Dietary BBR (50 mg/kg) alleviated the HCD-induced damaged intestinal barrier and excessive lipid deposition to inhibit inflammation and endoplasmic reticulum stress via the target AMPK/SREBP1 cascade in the intestine of largemouth bass [86]. Additionally, feeding BBR corrected intestinal permeability, inflammation, and imbalanced gut flora caused by HCD and HFD treatment, which greatly improved blunt snout bream growth performance [87]. Unfortunately, berberine lacks clinical trials regarding its use as a supplement for IBD patients.

**Table 1 molecules-28-06811-t001:** The details of experimental models, used doses, and the targets of Berberine in intestinal inflammation.

Chinese Herbal Compounds	Experimental Model	Used Dose	Targets	References
Berberine	3% DSS-induced mice	100 mg/kg	promote apoptosis in colonic macrophages; Tight Junction (ZO-1); MAPK (ERK1/2 and p38) pathway; NF-κB pathway	[75]
Berberine	1 μg/mL LPS-induced RAW 264.7	50 μM	Apoptosis; MAPK (ERK1/2, p38, and JNK) pathway; NF-κB pathway	[75]
Berberine	5% DSS-induced rat	10 mg/kg 40 mg/kg	P-gp/Nrf2	[76]
Berberine	100 nM Nrf2-siRNA Caco-2 cells	2.5 μM	P-gp/Nrf2	[76]
Berberine	LP CD4+ T cells from colitis SCID mice	100 µM	Th1/Th17-related JAK/STAT pathway; AMPK pathway	[78]
Berberine	5% DSS-induced cat	40 mg/kg 80 mg/kg	Tight Junction (ZO-1, ZO-2, E-cadherin, ZEB1, occludin E-cadherin, slug and N-cadherin); Gut Microbiota; TLR4/NF-κB Signaling Pathway; MTORC and Autophagy	[79]
Berberine	Gut microbiomes in stool samples collected from seven health volunteers	250 µM	Butyrate pathway	[80]
Berberine	3% DSS-induced mice	100 mg/kg	Tight Junction (ZO-1, E-cadherin, and occludin); The enteric nervous system (ENS); PI3K-AKT-mTOR; MAPK signaling pathway; NF-κB P65 signaling pathway; Pyroptosis	[77]
Berberine	100 ng/mL TNF-α-induced Caco-2/HT-29	12.5 μmol/L 25 μmol/L 50 μmol/L	Tight Junction (E-cadherin)	[77]
Berberine	One-day-old female yellow-feathered broilers	250 mg/kg	Cecal microbiota	[82]
Berberine	One-day-old Ross 308 broilers	1 g/kg 2 g/kg	Gut microbiota from the jejunum to the colon	[83]
Berberine	High carbohydrate diet (HCD)-induced largemouth bass (Micropterus salmoides)	50 mg/kg	Tight Junction (Occludin, Claudin-1, ZO-1, ZO-2); Apoptosis Endoplasmic reticulum stress; AMPK/SREBP1; Lipid metabolism	[86]
Berberine	High-fat diet (HFD)-induced and high carbohydrate diet (HCD)-induced blunt snout bream (Megalobrama amblycephala)	50 mg/kg	Fish intestinal barrier (physical, chemical, immunological and microbiological barriers)	[87]

### 3.2. Sanguinarine

Sanguinarine (SAN) is present in the Papaveraceae, Fumariaceae, Ranunculaceae, and Rutaceae plant families and in medicinal plants such as Sanguinaria canadensis (blood root), Poppy fumaria, Bocconia frutescens, Chelidonium majus, and Macleya cordata. It has been demonstrated to possess anti-inflammatory, anti-microbial, and anti-tumor properties [88,89,90]. It was reported that the biodistribution of 3H-labeled SAN was examined in rats after 96 h, and the majority of SAN was found in the gastrointestinal system, specifically the small intestine, colon, and their contents [91]. Sun et al. furnished evidence for the metabolism of SAN and revealed that SAN was converted into nontoxic DHSA in pig intestines [92].

SAN showed remarkably reverse results in the DAI, wet colon weight, body weight change, MPO activity, colon macroscopic damage index (CMDI), and colon histopathologic score (CHS) in acetic acid-caused UC mice at the doses (1, 5, and 10 mg/kg). SAN also depressed the secretion of proinflammatory cytokines in the serum and colon. Further, SAN reduced the inflammatory response by diminishing the induced upregulation of NF-κB p65 in the colon [93] (Table 2). It was also reported that SAN could alleviate indomethacin-treated small intestine injury by regulating the Nrf2/NF-κB pathways in rats. In vitro experiments have verified the targeting of Nrf2 by SAN, and Nrf2 knockdown led to the weakening of SAN protection and an increase in intestinal inflammation [94]. Moreover, SAN could suppress the high expression of NLRP3, caspase-1, and IL-1β in DSS-induced mice and LPS-stimulated THP-1 cells. Additionally, SAN distinctly influenced the abundance variations of various genera such as Mucispirillum, Escherichia-Shigella, Lachnospiraceae_NK4A136_group, and Helicobacter [95]. It has been proven that SAN played a preventive role in intestinal injury in mice and IEC-6 exposed to ionizing radiation by the downregulation of HMGB1/TLR4 pathway and the modulation of the disturbed intestinal flora and SCFAs, including acetic acid, butyric acid, valeric acid, and capric acid in colonic fecal samples [96]. Intriguingly, SAN in vitro trials reduced intestinal damage caused by trichinella infection in mice while also killing muscle larvae, adults, and newly borne larvae. SAN has a significant effect on the ratio of the intestinal villus to the crypt, the number of small intestine goblet cells, and ROS levels in the serum [97].

For animal production, plant-derived antimicrobial alkaloids are used as an alternative to antibiotics to promote the growth of economic animals. Recently, investigators have examined the effects of SAN supplementation on broilers. The dietary SAN supplementation enhanced mucosal morphology and decreased the concentrations of TNF-α and IL-4 in the jejunum. The cecal microbiota’s 16S rRNA gene sequence revealed that SAN significantly increased the ratio of Firmicutes/Bacteroidetes in the phylum and enriched the generas Lachnospiraceae and Ruminococcaceae [98], which was similar to the prophylactic effects of ceftiofur toward Enterobacteriaceae-infected early gastrointestinal diseases [99]. Further research revealed that SAN recovered broilers’ duodenum and ileum from necrotic enteritis (NE) infection, but that the jejunum showed no discernible changes. SAN upregulated acetic acid and butyric acid and downregulated NE-infected propinoic acid [100]. In addition, SAN improved average daily gains and average daily feed intake for early-weaned piglets. It was characterized that the quantity of Lactobacillus in the ileum and caecum was increased by the addition of SAN while decreasing the quantity of *Escherichia coli* and Salmonella spp. in the ileum, as well as Salmonella spp. in the caecum. SAN enhanced the amounts of acetate, butyrate, propionate, and total SCFAs to achieve better regulation of flora in the ileal and cecal contents [101]. Unfortunately, Sanguinarine lacks clinical trials regarding its use as a supplement for IBD patients.

**Table 2 molecules-28-06811-t002:** The details of experimental models, used doses and the targets of Sanguinarine in intestinal inflammation.

Chinese Herbal Compounds	Experimental Model	Used Dose	Targets	References
Sanguinarine	1 mL of 5% acetic acid in 0.9% saline instilled-Kunming mice	1 mg/kg 5 mg/kg 10 mg/kg	NF-κB p65	[93]
Sanguinarine	7.5 mg/kg indomethacin-induced SD rats	0.33 mg/kg 1 mg/kg 3.3 mg/kg	Nrf2/NF-κB pathways; Tight Junction (Claudin-1, ZO-1)	[94]
Sanguinarine	300 μmol/L-induced IEC-6 cells treated with Nrf2 siRNAs (50 nM)	0.25 μmol/L 0.5 μmol/L 1.0 μmol/L	Nrf2/NF-κB pathways; Tight Junction (ZO-1)	[94]
Sanguinarine	3% DSS C57BL/6 SPF mice	5 mg/kg 10 mg/kg	NLRP3-(Caspase-1)/IL-1β pathway; Gut Microbiota	[95]
Sanguinarine	0.5 μg/mL LPS-induced THP-1 cells	0.25 Μm 0.5 μM 1.0 μM	NLRP3-(Caspase-1)/IL-1β pathway	[95]
Sanguinarine	One-day-old female yellow-feathered broiler chickens	0.7 mg/kg of feed	Cecal microbiota	[98]
Sanguinarine	One-day-old Ross 308 broilers co-infected with Eimeria sp. and *C. perfringens* (type A strain) challenged at the rate of 4 × 10^8^ CFU	0.12 g/kg Sangrovit^®^	Cecal microbiota and SCFAs	[100]
Sanguinarine	healthy weaned piglets (Duroc × [LargeWhite × Landrace]) weaned at 21 days of age	50 mg/kg Sangrovit^®^	Intestinal microflora, SCFAs and ammonia in the ileum and caecum	[101]

### 3.3. Astragalus Polysaccharide

Astragalus membranaceous is one of the most popular herbal medicines in the world and is known as “Huangqi” in China. Astragalus polysaccharide (APS) is the most significant natural bioactive component in Astragalus membranaceus, consisting of glucose, arabinose, xylose, and mannose, and possesses multiple pharmacological properties, such as immune regulation, antioxidant activity, anti-tumor and anti-inflammation properties, among others [102,103].

Hu et al. performed a meta-analysis and concluded that APS treatment dramatically relieved colonic damage by reversing the levels of DAI, CMDI, CHS, TNF-α, IL-6, IL-1β, MPO, MDA, and SOD [104]. Pre- and post-addition of APS (100 μg/mL) could downregulate the expression of inflammation indicators and levels of short circuit current while increasing the mRNA expression of ZO-1 and occludin in LPS-infected Caco-2 cells [105] (Table 3). Chen et al. further researched APS anti-colitis effects on preventing ferroptosis through inhibiting the Nrf2/HO-1 pathway in RSL3-stimulated Caco-2 cells, as indicated by the reduction of the mRNA expression of ferroptosis-associated PTGS2, FTH, and FTL and the decrease in the levels of ferroptosis markers (MDA, GSH, and iron load) [106]. A previous study indicated that the immunoregulatory effects of APS were mediated by the activation of the NF-κB p65/MAPK signaling pathway [107].

In DSS-induced colitis, APS effectively inhibited the NLRP3 inflammasome in a dose-dependent manner, which acted to decrease the production of IL-1β and IL-18 [108]. Importantly, Liu et al. complemented the viewpoint that APS was essential for the response of the immune system, targeting modulation of the follicular helper T cells (Tfh)/regulatory T cells (Treg) balance, the emerging strategy of UC treatment. For this specific manifestation of the rise in Treg cells and associated nuclear transcription factor Foxp3, the reduction in Tfh cells and associated nuclear transcription factors Bcl-6 and Blimp-1, and the increase in Tfh cells. Meanwhile, APS prominently modulated the differentiation of Tfh subpopulations, with upregulated Tfh10 and Tfr and downregulated Tfh1, Tfh17, and Tfh21 in the peripheral blood of colitis mice after APS therapy [109].

The immunological characteristics of 2,4,6-trinitrobenzene sulfonic acid (TNBS)-induced colitis mice facilitate the study of the immune response as a factor in the pathogenesis of IBD. Due to APS’s excellent immunomodulatory ability, Gao explored the immune regulation of APS in the TNBS-induced colitis model and demonstrated that APS resulted in the upregulation of the GATA-3/T-bet ratio. T-bet and GATA-3 are TH1- and TH2-specific transcription factors and can control the differentiation of TH1/TH2 [110,111]. These results suggested that APS favored a shift to the TH2 phenotype and achieved anti-colitis [112]. It has also been reported that APS ameliorated the inflammatory response in TNBS-induced rats, probably by restoring the immune suppression of Treg cells and inhibiting IL-17 levels in Peyer’s patches (PPs) [113]. PPs modulate the intestinal inflammatory response and tolerance and situate abnormal expression in IBD patients [114]. Therefore, it might also be an immunomodulatory target of APS.

APS also protects the intestines of economic animals. After APS treatment in goslings infected with gosling plague, the changes in SOD, GSH-Px, and MDA revealed increased antioxidant levels in the jejunal. The decrease in the mRNA expression of IL-1β, IL-6, LITAF, NF-κB, COX-2, and PGE2, and IL-1β, IL-6, and TNF-α protein content embodied the anti-inflammation of APS. APS markedly increased serum IgG, IgM, C3, C4, and IFN-γ and improved organism immunity [115]. Feed supplementation of APS could improve the production performance and intestinal health in Chongren hens, attributed to Firmicutes and Lactobacteriaceae in the cecum microbiota [116]. The latest reports showed that APS alleviated intestinal inflammation challenged with necrotic enteritis (NE) in broilers, most likely through modifying the balance of Th17/Treg cells, the microbiota in the gut, and metabolites. Concretely, dietary APS supplementation took effect, manifesting a reduced proportion of Th17 cells and an increased proportion of Tc and Treg cells in the ileum. Reduction of the Th17/Treg ratio was a tendency to APS treatment. Furthermore, APS declined the number of *Clostridium perfringens* in the cecum and uric acid, L-arginine, and serotonin in the ileum and escalated α diversity, Rombousia, Halomonas, and cholic acid in the ileum, which was related to Th17/Treg balance and inflammation [117]. Additionally, APS pretreatment might efficiently prevent ducklings from contracting Muscovy duck reovirus (MDRV) through the activation of mucosal immune function [118]. Along with the development of APS application in multi-species, researchers found that it exerted its effects during growth and regulated immune and anti-inflammatory responses in Channa argus [119]. Unfortunately, APS lacks clinical trials regarding its use as a supplement for IBD patients.

**Table 3 molecules-28-06811-t003:** The details of experimental models, used doses and the targets of Astragalus polysaccharide in intestinal inflammation.

Chinese Herbal Compounds	Experimental Model	Used Dose	Targets	References
Astragalus polysaccharide	1 μg/mL LPS-induced Caco-2 cells	25 μg/mL 50 μg/mL 100 μg/mL 200 μg/mL	Inflammatory cytokines (TNF-α, IL-1β and IL-8); Tight Junction (Occludin, ZO-1)	[105]
Astragalus polysaccharide	3% DSS-induced mice	100 mg/kg 200 mg/kg 300 mg/kg	Ferroptosis (PTGS2, FTH, FTL, MDA, GSH, and iron load)	[106]
Astragalus polysaccharide	RSL3 (an inhibitor of glutathione peroxides 4)-induced Caco-2	25 μg/mL 50 μg/mL 100 μg/mL	NRF2/HO-1/Ferroptosis pathways	[106]
Astragalus polysaccharide	RAW264.7 cells pretreated with 5 μmol/L pyrrolidine dithiocarbamate (PDTC, NF-κB inhibitor)	25 μg/mL 50 μg/mL 100 μg/mL	NF-κB p65/MAPK signaling pathway	[107]
Astragalus polysaccharide	3% DSS-induced mice	100 mg/kg 200 mg/kg 500 mg/kg	NLRP3 Inflammasome	[108]
Astragalus polysaccharide	3% DSS-induced mice	200 mg/kg	Tfh/Treg cells	[109]
Astragalus polysaccharide	TNBS-induced rats	500 mg/kg	Th1/Th2 cells; GATA-binding protein-3/T-Box	[112]
Astragalus polysaccharide	TNBS-induced rats	400 mg/kg	Treg cells in Peyer’s patches	[113]
Astragalus polysaccharide	Goose parvovirus-infected seven-day-old healthy goslings	0.3 mL, purity 87.81%	Immunological ability (IgG, IgM, C3, C4 and INF-γ levels in serum and sIgA in the jejunum); Antioxidant function (GSH-Px, SOD and MDA); Inflammatory cytokines (IL-1β, IL-6, TNF-α, NF-κB, COX-2 and PGE2)	[115]
Astragalus polysaccharide	Chongren hens (240-day old)	100 mg/kg 200 mg/kg 400 mg/kg	Antioxidant function (CAT, T-AOC, SOD and MDA); Cecal microbiota	[116]
Astragalus polysaccharide	Arbor Acres broiler chicks (one day old) orally inoculated with 33,000 sporangia and *Clostridium perfringens* CVCC2030 at a concentration of 1 × 10^9^ CFU/mL	200 μg/mL	Th17/Treg cells; Gut Microbiota in ileum; Gut metabolome in ileum	[117]
Astragalus polysaccharide	One-day-old Muscovy ducklings with Muscovy duck reovirus	600 μg/mL	Mucosal immune function ((sIgA) and duodenal cytokine levels of IL-4, IL-6, IL-15 TNF-α, INF-γ)	[118]
Astragalus polysaccharide	Channa argus for two weeks old	250 mg/kg 500 mg/kg 1000 mg/kg 2000 mg/kg	Antioxidant function (SOD, CAT, MDA and GSH-Px); Immunological parameters (lysozyme, C3, C4 and IgM)	[119]

### 3.4. Curcumin

Curcumin originated in Curcuma longa L. and is reported to have anti-inflammatory, antioxidant, antimicrobial, and antitumor effects. Moreover, it acts on NF-κB, MAPKs, JAKs/STATs, TNF-α, IL-6, and many cellular other targets that effectively influence IBD progression. Therefore, curcumin is a promising candidate that possesses the ability to manage IBD [120].

Zhao et al. suggested that curcumin and its primary metabolites modulated the NF-κB/IκB pathway by inhibiting IκB degradation and NF-κB activation to reduce the inflammatory factors in LPS-stimulated RAW 264.7 macrophage cells [121] (Table 4). As assessed by DAI and histological injury scores, mice fed with DSS and given curcumin were significantly ameliorated. Deeply, curcumin suppressed the STAT3 pathway by decreasing phosphor-STAT3 activity and DNA-binding of STAT3 dimers [122]. Surprisingly, the first double-blind, placebo-controlled clinical experiment revealed that daily supplementation of 1500 mg of curcumin could improve clinical response and ameliorate life quality, as well as lower the serum levels of ESR and hs-CRP in UC patients [123]. IBD is considerably facilitated by the maturation of the inflammatory cytokines IL-1 and IL-18, which can be mediated by the NLRP3 inflammasome. Wu et al. found curcumin mechanistically restrained DSS-induced K+ efflux, cathepsin B release, and intracellular ROS production [124]. Some researchers argued that curcumin activated SIRT1/NRF2 and inhibited the TLR4 signaling pathway to improve necrotizing microscopic colitis and cell pyroptosis [125], as well as preventing DSS-induced excessive autophagy [126].

There was evidence demonstrating that curcumin restricted the amount and activation of CD8+ CD11c+ cells and improved the levels of anti-inflammatory substances, including IL-10, IFN-γ, and TNF-β1 [127]. The conversion of CD4+ T cells into Th1 cells is facilitated by CD11c+ DC activation, which releases a significant variety of inflammatory factors and intensifies inflammation [128]. In addition, reports suggested that CD8+ DCs were activated to generate specific CD8+ T cell immune responses [129], which required the involvement of the TLR3 signal [130]. Curcumin was able to reverse the activated state of naïve T cells induced by DSS, and its performance for increased frequency of CD45RA + CD62LCCR7+ cells, which biologically marked a quiescent state of naïve T cells. Further studies indicated that curcumin treatment led to an increase in the number of central memory T (TCM) cells while decreasing the number of CD4+ and CD8+ TCM cells. Additionally, it was evident that curcumin administration decreased the number of effector memory T (TEM) cells and their subsets, CD4+ TEM, in peripheral blood. These changes showed that curcumin had the ability to control how TCM cells, TEM cells, and their subpopulations differentiated. Liu et al. speculated that curcumin may interfere with the JAK-STAT pathway to regulate the differentiation of memory T cells for IBD [131]. Wei et al. provided supplementary instruction on the impact of curcumin on the regulation of memory B cell subsets, which showed the striking reduction of CD19 + CD27+ B cells and augment of CD19 + CD27+ IL-10+ and CD19 + CD27+ Tim-3+ B cells [132].

In addition to treating intestinal inflammation, curcumin also helps IBD patients with other problems. Curcumin significantly improved the symptoms of diabetes accompanied by UC via the target on Th17/Treg and restoration of the makeup of the gut microbiota [133]. IBD has a significant impact on mental health and can lead to anxiety and sadness. Xu et al. demonstrated that curcumin altered particular intestinal flora to enhance the phosphatidylcholine in the prefrontal cortex of DSS-induced mice and alleviated anxiety-like behaviors through the microbiota-gut-brain axis [134]. Interestingly, curcumin supplementation ameliorated aflatoxin B1-induced duodenal toxicity in chickens by downregulating CYP450 enzymes and inducing P-gp content [135]. Curcumin formulated in a solid dispersion significantly reduced *Salmonella enteritidis* (SE) recovery in the cecal tonsils of broiler chickens [136]. Solid dispersion of curcumin with polyvinylpyrrolidone (PVP) (1:9 ratio) increased the solubility and stability of curcumin, which was associated with an improvement in curcumin’s anti-SE action in vitro and in vivo trials. In addition, it was able to raise SOD activity while lowering intestinal permeability. The synergistic impact of further boric acid addition was used to reduce SE colonization in broilers’ intestines [137]. Additionally, the combination of copper acetate and curcumin (CA-CR) played overlapping and distinct roles in resisting the *Salmonella typhimurium* (ST) infection in broiler chickens. This composition dramatically reduced ST colonization in the cecal tonsils and maintained intestinal integrity. Importantly, CA-CR and CA, respectively, reversed the ST-induced decline in the genera Erysipelotrichaceae and Lachnospiraceae, which would positively affect the diversity and development of the gut microbiota [138].

Current work found that a hereon Ora-Curcumin-S specifically was delivered to the luminal side of the colon with minimal systemic absorption. Tummala et al. verified that this curcumin was significantly effective in improving colitis in the in vivo mice model via interfering with TLR4 signaling [139]. Moreover, newly-developed nanoparticle curcumin can be better absorbed, and it greatly improves experimental colitis via modulation of gut flora structure linked to regulatory properties of mucosal immune cells [140].

**Table 4 molecules-28-06811-t004:** The details of experimental models, therapeutic doses, and the targets of Curcumin in intestinal inflammation.

Chinese Herbal Compounds	Experimental Model	Used Dose	Targets	References
Curcumin and its metabolites (tetrahydrocurcumin, hexahydrocurcumin and octahydrocurcumin)	1μg/mL LPS-induced RAW 264.7	3.125 µM 6.25 µM 12.5 µM 25 µM 50 µM 100 µM	NF-κB/IκB-α	[121]
Curcumin	5% DSS-induced mice	50 mg/kg	STAT3 pathway	[122]
Curcumin	UC patients	1500 mg	The serum hs-CRP and ESR levels	[123]
Curcumin	Bone marrow derived macrophages (BMDMs) pretreated with 1 μg/mL LPS for 3 h and stimulated with 3% DSS for 24 h	10 Μm 25 μM 50 μM	K+ efflux, intracellular ROS, cathepsin B and NLRP3 inflammasome	[124]
Curcumin	3% DSS-induced mice	100 mg/kg	NLRP3 inflammasome	[124]
Curcumin	The rat NEC model was established by hypoxia (treated at 5% O2 for 10 min) and hypothermia (treated at 8 °C for 10 min) stimulation	20 mg/kg 50 mg/kg	SIRT1/NRF2 pathway; TLR4 pathway	[125]
Curcumin	5% DSS-induced mice	15 mg/kg 30 mg/kg 60 mg/kg	Autophagy (ATG5, LC-3II), beclin-1, bcl-2)	[126]
Curcumin	100 mg/kg TNBS-induced mice	100 mg/kg	CD8 + CD11c+ cells	[127]
Curcumin	3% DSS-induced mice	100 mg/kg	The differentiation of naïve, TCM, and TEM cells in the peripheral blood; JAK1/STAT5 signaling	[131]
Curcumin	BALB/c male mice were freely given a 3% DSS for 7 days and 2% DSS for 7 days	100 mg/kg	Memory B cells; Bcl-6-Syk-BLNK signaling	[132]
Curcumin	1.5% (*w/v*) DSS-induced type 2 diabetic mice	100 mg/kg	Th17/Treg cells; Colonic microbiota	[133]
Curcumin	3% DSS-induced mice	100 mg/kg	Microbial-Brain-Gut Axis; Fecal microbiota; Lipid metabolism	[134]
Curcumin	One-day-old Arbor Acres broilers with 5 mg AFB1/kg feed	150 mg 300 mg 450 mg	P-glycoprotein (P-gp); CYP450 enzymes	[135]
Curcumin formulated in a solid dispersion (SD-CUR)	*S. enteritidis* (SE) infection in broiler chickens	0.1% (1 kg/Ton of feed)	Superoxide dismutase (SOD); total intestinal IgA	[136]
Solid dispersion of curcumin with polyvinylpyrrolidone and boric acid	Day-of-hatch male Cobb-Vantress broiler chickens	Basal diet plus 1% BA-CUR/PVP	Superoxide dismutase (SOD); total intestinal IgA	[137]
Combination of curcumin and copper acetate	Day-of-hatch male Cobb-Vantress broiler chickens challenged with 10^4^ CFU of *Salmonella typhimurium*	Basal diet supplemented with 250 mg/kg of copper (II) acetate hydrate and 0.2% curcumin	Cecal microbiota	[138]
Ora-Curcumin-S prepared by molecular complexation of curcumin with a hydrophilic polymer Eudragit^®^ S100	2.5% DSS-induced mice	15 mg/kg	NF-κB activation	[139]
Ora-Curcumin-S prepared by molecular complexation of curcumin with a hydrophilic polymer Eudragit^®^ S100	DC2.4 cells stimulated with MPLA (2 µg/mL) or *E. coli* (5.0 × 10^5^/mL)	5 µg/mL 10 µg/mL	TLR-4 signaling	[139]
Nanoparticle curcumin	3% DSS-induced mice	A normal rodent diet (containing 0.2% (*w/w*) nanoparticle curcumin)	NF-κB activation; CD4 + Foxp3+ regulatory T cells and CD103+ CD8α− regulatory dendritic cells; The fecal short-chain fatty acid (acetate, butyrate and propionate); Gut microbiota	[140]
Nanoparticle curcumin	100 ng/mL TNF-αinduced HT-29 cells	10 μM	NF-κB activation	[140]

### 3.5. Cannabinoids

The term cannabinoid originally restrictedly referred to the phytocannabinoids of Cannabis sativa L. in cannabis with the typical C21 structure and their transformation products. Today, the term cannabinoid can encompass all ligands and relevant compounds of the cannabinoid receptor, including endogenous ligands of the receptor and a large variety of synthetic cannabinoid analogs [141]. Mono- and sesquiterpenes, sugars, hydrocarbons, steroids, flavonoids, nitrogenous chemicals, and amino acids are all included in the class of substances known as cannabinoids. A total of 66 phytocannabinoids have been identified, most of them belonging to several subclasses or types: cannabichromene (CBC), cannabidiol (CBD), cannabinodiol (CBDL), cannabielsoin (CBE), cannabigerol (CBG), cannabicyclol (CBL), cannabinol (CBN), cannabitriol (CBTL), Δ8-THC and Δ9-THC types [142]. Tetrahydrocannabinol (9-THC) is the most prevalent cannabinoid and is primarily responsible for giving cannabis its euphoric effects. CBD is the second most prevalent cannabinoid. Dronabinol pills and smoking are the two most frequent ways to consume cannabis products. The rectal route with suppositories, cutaneous, and sublingual delivery are being studied for therapeutic applications. Since CBDs are fat-soluble, they easily pass across cell membranes, the blood–brain barrier, and the placenta. After entering the human body, CBDs will work in tandem with plasma proteins to carry out their pharmacological or hazardous effects [143].

In the gastrointestinal tract, the CB1 receptor (CBR1) is mainly located in myenteric, submucosal neurons, and nonneuronal. The CB2 receptor (CBR2) is mainly located on inflammatory and epithelial cells, as well as myenteric and submucosal neurons [144]. To date, two cannabinoid receptors have been identified, CBR1 and CBR2, both activated G proteins (Gi proteins) inhibitory to adenylate cyclase (AC), thus inhibiting the conversion of ATP to cyclic AMP (cAMP), and both are positively coupled to MAPK signaling [145]. CB2R signaling mechanisms involved the activation of the PI3K-AKT pathway and increased the synthesis of the sphingolipid messenger ceramide, thus having a pro-survival and a pro-apoptotic effect, respectively [146]. CB1 and CB2 receptors express immune cells and are thought to play a role in immune homeostasis and control. Although the majority of studies showed that the administration of cannabinoids had inhibitory effects on immune cells, a number of recent studies have demonstrated that endocannabinoids might have some stimulatory impact on the immune system. The mechanism by which cannabinoids activate or inhibit immune cells is not yet clear. CB1 and CB2 mRNA expression are present in human and mouse immune cells in the order B cells > natural killer cells > monocytes > neutrophils > CD8 leukocytes > CD4 leukocytes [147]. The pattern of distribution showed wide variations in the mRNA levels in the main human blood cells. For CB1, the order of the levels in B cells > natural killer cells > polymorphonuclear neutrophils > T8 cells > monocytes > T4 cells, and for CB2, the order of the levels was the same for immune cells [148]. After LPS stimulation of splenocytes, CB2 mRNA expression was downregulated in a dose–response manner, whereas in contrast, anti-CD40 antibody co-stimulation upregulated CB2 expression, which could be reversed by IL-4 [149]. Additionally, Noe et al. found that splenocytes stimulated with T cell mitogens showed a decrease in the CB1 message, while cultured stimulated with B cell mitogens showed an increase in the message. These confirmed the differences in CB1 expression in response to different stimuli from different immune cells [150].

Kozela, E. et al. used the BV-2 microglial cell line to assess the effects of Δ9-THC and CBD on the LPS-activated microglial secretion of proinflammatory cytokines such as IL-6, IL-1β, and IFNβ. They found that although both Δ9-THC and CBD exerted anti-inflammatory effects on the production of proinflammatory cytokines in activated BV-2 cells, Δ9-THC and CBD acted through different, partially overlapping mechanisms. Both THC and CBD decrease the activation of proinflammatory signaling by interfering with the TRIF/IFNβ/STAT pathway. Furthermore, CBD additionally suppressed the activity of the NF-κB pathway and potentiated an anti-inflammatory negative feedback process via STAT3 [151] (Table 5). Do et al. found the addition of Δ9-THC and anandamide, an endogenous cannabinoid, would make DCs turn to apoptosis. Treatment with Δ9-THC induced the cleavage of caspase-2, -8, -9, and Bid, decreased mitochondrial membrane potential, and cytochrome c release, suggesting the involvement of apoptosis and mitochondrial pathways. Importantly, the same degree of apoptosis occurred, suggesting that Bid cleavage was not a critical step for induction of apoptosis in DCs. Additionally, there was no activation of p44/p42 MAPK, p38 MAPK, or stress-activated protein/JNK pathway in THC-treated DCs. However, Δ9-THC induced the NF-κB pathway via increased phosphorylation of IκB-α and transcription of several apoptotic genes regulated by NF-κB, which led to the DCs depletion to suppress the immune response [152]. The mechanism of action of immune suppression by cannabinoids involves suppression of IL-2 production in PMA/Io-stimulated splenocytes via inhibition of ERK MAPK activation [153].

IL-17A is a cytokine linked to IBD, where elevated expression is seen in inflamed colonic tissue compared to controls. The study showed that IL-17A could induce epithelial and crypt damage in human colonic mucosal explant tissue without directly influencing epithelial permeability in Caco-2 cells over a 46 h incubation period. Interestingly, the cannabinoid ligands (cannabinoids, anandamide, cannabidiol) attenuated IL-17A-mediated mucosal damage, providing evidence that endocannabinoids or phytocannabinoids may offer mucosal protection and immune regulation [154]. IFNγ and TNF-α stimulated Caco-2 cells were used as an in vitro intestinal epithelial model system in inflammatory conditions. Surprisingly, Alhamoruni, A. et al. used this model to find that endocannabinoids would further worsen the increased permeability, while phytocannabinoids or CB1 receptor antagonism speeded the recovery of permeability in inflammatory conditions. It was hypothesized that endocannabinoids may play a role in gut permeability during inflammation through CB1 receptors [155]. The study confirmed that under inflammatory conditions, PEA and CBD suppressed the phosphorylation of several intracellular proteins (e.g., CREB, JNK, NF-κB, ERK1/2, Akt, STAT3, and STAT5) in Caco-2 stimulated with IFNγ and TNF-α, whereas these did not suppress the secretion of pro-inflammatory cytokines (IL-6, IL-8, and IL-17). Reversely, in explant human colonic tissue stimulated with IFNγ and TNFα, PEA and CBD both suppressed the phosphorylation of intracellular proteins and also prevented the increased secretion of pro-inflammatory cytokines [156]. One of the recent prospective studies focused on 21 Crohn’s patients, who were given cannabis two times each day in the form of cannabis flowers or tetrahydrocannabinol for eight weeks. The results showed that in a given 11.5 mg tetrahydrocannabinol group, about 90% of the patients had varying degrees of improvement and had better appetite and sleep [157].

**Table 5 molecules-28-06811-t005:** The details of experimental models, therapeutic doses and the targets of Cannabinoids in intestinal inflammation.

Chinese Herbal Compounds	Experimental Model	Used Dose	Targets	References
Δ9-tetrahydrocannabinol (THC)	100 ng/mL LPS-induced BV-2	1 μM 5 μM 10 μM	TRIF/IFNβ/STAT1/STAT3 pathway	[151]
Cannabidiol (CBD)	100 ng/mL LPS-induced BV-2	1 μM 5 μM 10 μM	IRAK-1/NF-κB/IκB; TRIF/IFNβ/STAT1/STAT3 pathway	[151]
Δ9-tetrahydrocannabinol (THC)	Mature dendritic cell (DC) from Bid-knockout (KO) mice	5 μM	NF-κB/IκB; Caspase-2/Caspase-8/Cytochrome c/Apoptosis	[152]
Endocannabinoids	Mature dendritic cell (DC) from Bid-knockout (KO) mice	20 μM	Apoptosis	[152]
Cannabinol (CBN) and WIN-55212-2 (a synthetic cannabinoid)	Splenocytes activated by PMA/Io (40 nM/0.5 μM) for 4 h	1 μM 10 μM 20 μM	AP-1 DNA binding; ERK MAPK/IL-2	[153]
Anandamide (AEA) and cannabidiol	IL-17A (10 ng/mL)-induced human colonic explant tissue	10 μM	IL-17A/matrix metalloprotease activity/mucosal damage	[154]
Δ9-tetrahydrocannabinol (THC), endocannabinoids anandamide (AEA) and 2-arachidonylglycerol (2-AG)	Caco-2 cells treated with IFNγ and TNFα (10 ng/mL) for 24 h	10 μM	CB1 receptor/intestinal permeability	[155]
Cannabidiol (CBD)	Caco-2 cells treated with IFNγ and TNFα (10 ng/mL)	10 μM	the phosphorylation level of CREB, NF-κB, JNK and STAT5	[156]
Palmitoylethanolamide (PEA)	Caco-2 cells treated with IFNγ and TNFα (10 ng/mL)	10 μM	the phosphorylation level of CREB and JNK	[156]
Cannabidiol (CBD)	Human colonic explants treated with IFNγ and TNFα (10 ng/mL)	10 μM	the phosphorylation level of NF-κB, Akt, p70S6K, STAT3 and STAT5	[156]
Palmitoylethanolamide (PEA)	Human colonic explants treated with IFNγ and TNFα (10 ng/mL)	10 μM	phosphorylation level of STAT3	[156]

## 4. Chinese Herbal Formulae for Intestinal Inflammation

In Chinese medicinal theory, IBD is characterized as “Dysentery”. The weakened “Qi” of the spleen, the digestive function indicator, and a dysregulated mixture of “Heat” and “Dampness” induce an imbalance of intestinal homeostasis, which subsequently causes symptoms of IBD. Numerous Chinese herbal formulae have been developed over thousands of years and are utilized to treat a wide range of diseases. Owing to the synergistic usage of herbs, the efficacy and mechanism of action of herbal compounds and their constituent herbs in the management of IBD have been extensively reported in the literature [158].

### 4.1. Wumei Wan (WMW)

WMW from the Treatise on Febrile Diseases is a classic Chinese herbal medicine formula that has been tested for the treatment of UC. The ten herbs that make up WMW include Prunus Mume, Rhizoma coptidis, and Cortex Phellodendri. WMW has a long history of clinical use in China for the management of persistent dysentery and chronic diarrhea, and it has proven to be highly effective. The most frequent and severe symptoms in colitis patients are diarrhea and abdominal pain. Therefore, WMW has anti-diarrheal and analgesic properties. Fructus mume is the main ingredient of WMW, and it is reported that it has the effect of reducing diarrhea and enhancing digestive function [159]. WMW contains analgesic properties in the form of Radix aconiti lateralis preparata, Rhizoma zingiberis, Ramulus cinnamomi, Herba Asar, and Pericarpium zanthoxyli. Aconitine is the primary bioactive component of WMW among them. Aconitine possesses pharmacological actions such as anti-cancer, anti-inflammatory, immunomodulatory, and analgesic activities, according to scientific pharmacological study. According to the hypothesis from traditional Chinese medicine, aconitine’s anti-inflammatory and immunomodulatory properties may be related to its “tonifying fire” and “helping Yang” properties, while its analgesic properties may be connected to its “eliminating cold” to halt pain properties [160]. In addition, the State Food and Drug Administration of China approved WMW in 2002 for the management of gastrointestinal disorders.

According to Liu et al., WMW had superior effects than dexamethasone in preventing diarrhea, colon weight gain, colonic accretion, ulceration, and an increase in MPO activity in TNBS-induced rats. In order to promote the balance of Th1/Th2 cells in rats with colitis, WMW increased IFN-γ and decreased IL-4 to address the immunological imbalance in colitis. Contrary to dexamethasone, WMW reversed the elevated G−/G+ levels in colitis for the balance of the intestinal flora (Figure 2) [161]. Additional experiments by Wang et al. showed that Jiaweiwumei decoction combined with Ruxiang and Moyao could successfully prevent the colon injury caused by TNBS through enhanced IL-10 in serum and intestinal mucosa as well as an increased percentage of CD4+/CD25+ T regulatory cells in peripheral blood. This might lessen the immune reaction and restore the integrity of the intestinal mucosa [162]. WMW prevented intestinal fibrosis in the fibrotic state of colitis, as seen by the decline in various fibrotic markers, such as α-SMA, collagen I, MMP-9, and fibronectin. Colon fibroblast proliferation and EMT, two crucial mediators during fibrosis, were inhibited after receiving WMW. Importantly, several key profibrotic pathways, including TGF-β/Smad and Wnt/β-catenin pathways, were downregulated by WMW treatment [163]. WMW alleviated DSS-induced colon shortening, decreased histological scores in mice, and downregulated the mRNA expression and protein content of proinflammatory factors TNF-α, IL-1α, IL-1β, and so on. According to data from network analysis and KEGG enrichment, the AGE-RAGE signaling pathway was substantially correlated with WMW-treated UC and would be the pivotal target involved in the anti-UC effect of WMW. It was detected that taxifolin, rutaecarpine, kaempferol, quercetin, and luteolin were found to be potential active components for the anti-colitis activity of WMW, and it effectively decreased the level of RAGE protein to inhibit the activation of NF-κB and enhance the colonic ZO-1 expression, therefore intestinal barrier integrity in DSS-induced mice [164]. Recent studies demonstrated that necroptosis, a recently identified programmed necrosis, is crucial to the pathogenesis of IBD. It has been proven that elevated necroptosis was strongly correlated with IBD intestinal inflammation and augmented the disease’s manifestation. Inhibiting RIPK3 activity has been extensively explored by many researchers as a direction for suppressing necroptosis and proved effective; furthermore, boosting RIPK3 O-GlcNAcylation can diminish RIPK3 activation, thereby inhibiting necroptosis and ultimately relieving colitis [165]. As the key enzymes in O-GlcNAcylation, O-GlcNAc transferase (OGT), and O-GlcNAcase (OGA), Wu et al. showed that WMW increased colonic O-GlcNAcylation levels to inhibit necroptosis and alleviate TNBS-induced colitis, as demonstrated by the elevated OGT activity and decreased OGA activity in TNBS-induced colitis in mice. There were a total of 11 maker compounds in WMW that were identified by high-performance liquid chromatography (HPLC), including citric acid, phellodendrine, ferulic acid, coptisine, jatrorrhizine, berberine, hesperidin, cinnamaldehyde, aconitine, ginsenoside Rb1, and 6-gingerol. Based on the results of molecular docking, hesperidin, coptisine and ginsenoside Rb1 may exert a major role in the regulation on OGT and OGA activities by WMW [166]. Additionally, microvascular permeability (mVP), perivascular edema, and epithelial hypoxia precede epithelial barrier dysfunction, which progresses to erosions, ulceration, and inflammation [167]. Permeability changes are also closely related with UC pathophysiology. WMW could reduce the typical UC symptoms brought on by acetic acid, repair damaged colon tissue through inflammatory responses, and control mVP through the VEGF-PI3K/AKT-eNOS signaling pathway [168].

### 4.2. Gegen–Qinlian Decoction (GQD)

GQD is a traditional Chinese medicine prescription composed by Puerariae Lobatae Radix, Scutellariae Radix, Coptidis Rhizoma, and Glycyrrhizae Radix et Rhizoma Praeparata cum Melle, which has the formula 8:3:3:2. Diarrhea with fever is traditionally and medically treated as both an “external and internal symptom” of GQD. The dried root of Pueraria lobata (Willd.) Ohwi is known as Puerariae Lobatae Radix, and it mostly includes isoflavones and isoflavone glycosides, including formononetin, daidzin, and daidzein. The dry root of Scutellaria baicalensis Georgi is known as Scutellariae Radix, and it mostly contains flavonoids such as baicalin, baicalein, wogonoside, and wogonin. The dried rhizome of Coptis chinensis Franch., Coptis deltoidea C. Y. Cheng et Hsiao, or Coptis teeta Wall is known as a “Coptidis Rhizoma,” and it mostly includes alkaloids such as berberine, jatrorrhizine, coptisine, and palmatine. Glycyrrhizae Radix et Rhizoma Praeparata cum Melle is the dry root and rhizome of Glycyrrhiza uralensis Fisch., Glycyrrhiza inflata Bat, or Glycyrrhiza glabra L.; it mainly contains flavonoid glycosides, triterpenoid saponins, and free phenolic compounds, such as glycyrrhizic acid, glycyrrhetinic acid, liquiritin, and isoliquiritin. The chemical components of GQD, however, cannot be viewed as a straightforward superposition of the chemical components of the four distinct TCMs [169]. Based on quantitative studies, natural compounds in GQD can be grouped into eight groups according to their structures: flavonoid C-glycosides, flavonoid O-glucuronides, benzylisoquinoline alkaloids, free flavonoids, flavonoid O-glycosides, coumarins, triterpenoid saponins, and other atypical and abundant backbones [170]. The tissue distribution pattern of the main active ingredients of orally administered GQD in mice, particularly in the liver and colon, was made known. In contrast to the colons, where puerarin, berberine, epiberberine, coptisine, palmatine, jatrorrhizine, and baicalin were enriched, the livers exhibited comparatively high exposure levels of daidzein, baicalin, and baicalein. According to a publication, oral administration of 10 and 50 mg/kg puerarin alleviated DSS-induced colitis due to the downregulation of the NF-κB pathway and upregulation of the Nrf2 pathway [171]. Integrated pharmacokinetic studies found that “integrated constituents” of GQD achieved their peak concentration at 4 h in the portal vein, the systemic circulation, the livers, and the colons, where they also accumulated. Furthermore, the concentration of the “integrated constituents” in the colons was roughly ten times greater than that in the livers, both of which were significantly higher than that in the systemic circulation, indicating that inflammatory intestinal diseases are probably the primary indicator of oral GQD [172]. The intestinal transit of GQD was examined using the Caco-2 cell monolayer model, and the findings showed that the alkaloids, flavonoid C- and O-glycosides, O-glucuronides, and coumarin showed good permeability. The absorption of zylisoquinoline alkaloids in Huang-Lian, flavonoid C-glycosides in Ge-Gen, and coumarins and flavonoid O-glycosides in Gan-Cao could all be significantly improved by GQD [173].

It has been reported that GQD is effective against inflammatory intestinal reactions brought on by diarrhea, UC, and other conditions [169]. Some scholars demonstrated the efficacy of GQD in treating colitis, as evidenced by the decreased CMDI and CHS, the lower colonic wet weight, as well as increased SOD activity and decreased levels of MDA and inducible nitric oxide synthase in TNBS-induced colitis. This treatment inhibited the production of inflammatory factors (i.e., TNF-α, IL-1β, cyclooxygenase-2, macrophage inflammatory protein-2, and intercellular adhesion molecule-1). In addition, it was shown that GQD was able to suppress the activation of TLR2, TLR4, and NF-κB p65. The data suggested that GQD exhibited protective benefits in UC via modulating the balance of oxidants and anti-oxidants and inhibiting the TLR/NF-κB signaling pathway (Figure 3) [174]. Li et al. also verified that GQD possessed the anti-UC pesticide effect in the DSS-induced model via achieving anti-oxidant and reducing the inflammatory factors, and further found that GQD administration significantly reduced the infiltration of Th17 and Treg cells into the colon, peyer’s patches, and spleen, as well as maintaining the balance between two types of cells. Unbalanced Th1/Th2 cells then went into equilibrium under GQD control. The reason for this was the role of GQD in modulating the homeostasis of infiltrating immune cells in a DSS-stimulated colon was achieved by inhibiting the JAK2/STAT3 pathway, which is associated with the differentiation of CD4+ T cells [175]. Tang et al. made a compliment about the Treg/Th17 balance regulated by GQD and suggested that QGD intervention boosted the abundance of gut flora-producing SCFAs, which influenced the intestinal immune responses to improve colitis. Concretely, the FMT experiment confirmed that GQD-treated fecal bacteria was transferred, generating improved diversity and discordance of the gut microbiota in DSS-induced colitis, and GQD treatment promoted the frequency of Treg cells and downregulated the ratio of Th17 to maintain intestinal homeostasis [176]. The differentiation and homeostasis of the intestinal mucosa are both controlled by notch signaling [177]. In both acute and chronic UC models, GQD could repair the damaged colonic mucosa. The study found that Notch signaling was hyperactive in acute UC mice and hypoactive in chronic UC mice. Surprisingly, GQD downregulated the Notch signaling pathway in the acute UC model induced by 3% (*w/v*) DSS for seven days to promote the differentiation of goblet cells and the release of more protective peptides and upregulated the Notch signaling pathway in the chronic UC model induced by 2% (*w/v*) DSS for three cycles to promote the proliferation of the epithelium and the differentiation of absorption cell lines. Li et al. proposed that GQD controlled the TLR4 signaling pathway with the downstream component TNF-α in order to alter the disordered and unbalanced Notch pathway without affecting the regular Notch pathway [178]. Tang et al. found that the mechanisms by which MGQD protects against DSS-induced chronic colitis may involve regenerating goblet cells, mending the intestinal mucus barrier, and preventing NLRP3/IL-1β signaling pathway activation, which subsequently reduces the activation of γδT17 cells [179]. γδT cells are highly motile and can cover a large area of tissue by occluding and causing cell-cell interaction as they move through the intestinal epithelium. γδT cells could enhance antimicrobial peptide release and contribute to intestinal epithelial repair, acting as a bidirectional regulatory role of immunological homeostasis in the intestinal mucosa [180]. Irinotecan (CPT-11), a water-soluble semi-synthetic derivative of camptothecin that has been given worldwide approval for the treatment of colorectal cancer patients, is well recognized to cause severe gastrointestinal damage in patients during the early stages of clinical research, particularly delayed diarrhea [181]. Zhang et al. revealed that the obtained GQD extract by macroporous resin elution contained puerarin, liquiritin, berberine, and baicalin at concentrations of 27.2 mg/g, 4.6 mg/g, 491.4 mg/g, and 304.2 mg/g, respectively. These compounds had a protective effect on reducing diarrhea in mice after CPT-11 administration. The manifestations included reduction of inflammatory factors (IL-1β, COX-2, ICAM-1, and TNF-α), anti-oxidative stress (Keap1/Nrf2 pathway), and repair of the intestinal barrier (ZO-1, HO-1, occludin). Additionally, utilizing a colon tumor xenograft model, it was found that GQD extracts improved synergistically with CPT-11 in the prevention of colonic tumor growth, outperforming both medications by themselves [182]. Using network pharmacology and animal experimental validation, GQD was uncovered to have a therapeutic effect against UC via downregulating the EGFR/PI3K/AKT signaling pathway in UC mice treated with 3% DSS and the PI3K/AKT/NF-κB signaling pathway in diarrhea mice treated with CPT-11 [183,184]. Based on the 16Sr DNA method and targeted metabonomics, Jiawei GQD with puerarin, baicalein, berberine, glycyrrhizic acid, and magnolol, could not only manage the gut microbiota by raising Akkermansia and Romboutsia but also by lowing Escherichia-Shigella. More significantly, J Jiawei GQD improved colonic inflammation in UC rat models in both the general environment and the Lingnan region by increasing the production of propionate and total short-chain fatty acids (SCFAs) in colitis rats, regulating medium and long-chain fatty acids (M-LCFAs), bile acids (BAs) homeostasis, and amino acids (AAs) metabolism [185]. After a systematic review and meta-analysis, they concluded that GQD was significantly effective in pediatric diarrhea and recommended that it should be applied for a period of greater than five days along with unaltered compounded formulas [186].

Importantly, GQD could treat cecum lesions and bloody diarrhea in chickens infected with Eimeria tenella to improve growth performance via the targets SRC, STAT, and PPARG [187]. In a comparative pharmacokinetic study, the absorption of Puerarin, Wogonin, and Daidzein increased in the *E. coli* infective diarrhea minipig model compared to the control group. It was shown that the pharmacokinetic properties of normal and bacterial diarrhea minipig model animals for the same treatment differed, which was more in line with the pathophysiology and improved drug application [188]. Interaction between the intestinal flora may explain the aforementioned events. Tan et al. confirmed that the microbial metabolism of GQD components by gut flora in the diarrheal group was faster than that in the normal group. Furthermore, the ingredients of GQD were metabolized by β-glucosidase, β-glucuronidase, and nitroreductase, and their activities were higher during a diarrheal state than they would have been under normal circumstances. It may be related to the rise in Escherichia, and this modification might improve GQD’s effectiveness [189]. GQD exerts its pharmacological effects on diarrhea piglets given with *E. coli*, such as a decline in watery stools, fecal water content, colonic mucosal injury, and histological score in GQD-treated pigs. Moreover, after treatment with GQD, there was a rise in the relative abundances of bacteria, including Akkermansia, Bacteroides, Butyricimonas, Clostridium, Ruminococcus, and Phascolarctobacterium. These bacteria have been recognized as important commensal bacteria that can produce SCFAs and prevent diarrhea. According to these bacteria, GQD treatment did indeed raise the levels of SCFAs like butyric acid, propionic acid, and acetic acid. The elevated amounts of SCFAs suppressed histone deacetylase and the NF-κB pathway and probably further helped to attenuate mucosal inflammation and diarrhea [190]. Zheng et al. showed that GQD therapy effectively inhibited pathogenic *E. coli* and promoted the growth of Lactobacillus in the diarrheal piglet model given by *E. coli* and based on bacterial culture in vitro. GQD may treat diarrhea and repair the intestinal mucosal barrier by facilitating Lactobacillus growth and inhibiting the TLR2/MyD88/NF-κB signaling pathway [191].

### 4.3. Banxia Xiexin Decoction (BXD)

BXD, a representative Chinese herbal formula, was originally described in the Treatise on Febrile Diseases (Shang Han Lun), formulated by Zhongjing Zhang. The BXD is composed of seven herbs (i.e., Rhizoma Pinelliae, Radix Scutellariae, Rhizoma Coptidis, Radix et Rhizoma Ginseng, Rhizoma Zingiberis and so on). In TCM theory, BXD’s formulation embodies the balance of acrid-opening, bitter-down-bearing, and sweet-tonifying properties. These properties are based on seven typical crude herbs and their bioactive ingredients. By combining various types of herbs, the stringent “sovereign, minister, assistant, and courier” principle, which was adopted from “Huangdi’s Internal Classic,” was followed to increase the effectiveness of TCM and decrease toxicities or negative effects. Coptidis Rhizoma is the “sovereign” in BXD and is used to treat the primary illness. Scutellariae Radix serves as the “minister” to strengthen the effects of the “sovereign.” To lessen the potential toxicity of the “minister” or “sovereign” and alleviate any accompanying symptoms, Pinelliae Rhizoma and Zingiberis Rhizoma serve as the “assistant” plants. The “courier” herbs that direct the other herbs to the target organs are Ginseng Radix et Rhizoma, Glycyrrhizae Radix et Rhizoma Praeparata Cum Mell, and Jujubae Fructus. Additionally, it’s possible that the advantageous effects of these active components from various herbs are coordinated and harmonized to produce overall responses that allow BXD to simultaneously affect several targets and exert synergistic therapeutic effects (Figure 4) [192]. Baicalin, wogonoside, berberine, coptisine, and liquiritin were shown to be the primary active substances in BXD among the test flavonoids, alkaloids, and saponins, according to the absolute quantification of bioactive components in BXD. Additionally, certain minor but significant components were found and concurrently identified, including oroxylin A, liquiritigenin, isoliquiritin, and isoliquiritigenin [193].

It is utilized to treat gastroenteritis, UC, and diarrhea [194,195]. Chen et al. revealed that treatment with BXD significantly attenuated experimental colitis in DSS-induced mice by ameliorating body weight, DAI, colon shortening, and colonic histology score. The curative mechanism of BXD is involved in the inhibition of NF-κB p65 phosphorylation and the increase in Nrf2 content. BXD may produce anti-inflammatory effects by suppressing oxidative stress in the colorectum [196]. Zhang et al. found that therapy with BXD caused UC rats given 7% DSS to experience significant remission, with better DAI scores, a longer colon, reduced loss of epithelial crypts, breakdown of the mucosal barrier, and infiltration by inflammatory cells. Further study indicated that BXD suppressed Akt/IκBα kinase activity through NF-κB activation, blocking ERK expression and subsequently suppressing NF-κB p65 activation, which notably reduced the production of inflammatory factors including TNF-α, IL-1β, IL-6, and IL-8. More importantly, the proteins upstream and downstream of the Akt/MAPK signaling pathway prevent the tissue damage caused by inflammatory cytokines by preventing cellular signals from being amplified and by efficiently suppressing additional signal transmission [197]. Interestingly, the different decoction methods can bring the difference in the pharmacological effects of traditional herbal prescriptions. According to Zhang’s research, the contents of the effective components (quercetin, baicalein, wogonin, and baicalin) in BXD extracts with the ancient extraction method (BXD-AED) were higher than those in BXD extracts with the modern extraction method (BXD-MED), and the BXD-AED performed better in alleviating UC severity compared with the BXD-MED at similar doses [198]. Using a systems pharmacology approach and molecular docking, the researchers investigated potential pathways for BXD treatment in the fight against irritable bowel syndrome (IBS) and predicted promising drug targets, such as PTGS2, NOS2, and PRSS1. Therefore, BXD may be a candidate medicine to facilitate the development of IBS therapy [199]. It is well known that irinotecan (CPT-11), which is used for the first-line treatment of metastatic colorectal cancer, causes severely delayed-onset diarrhea. For this side effect, BXD could alleviate the gastrointestinal symptoms stimulated by CPT-11, and it was pointed out that active components against CPT-11-induced intestinal toxicity included coptisine, baicalin, berberine, and palmatin [200]. Surprisingly, Chinese herbs exert the therapeutic ability to treat different diseases in the same way, underlying their multi-pathway and multi-target pharmacological effects. Huang et al. suggested that BXD might achieve the effect of the same treatment for depression and UC by primarily focusing on the 5-hydroxytryptamine (5-HT) synaptic, HIF-1, and NF-κB signaling pathways, and the effective components lied in eight key proteins such as IL-6, MAPK1, STAT3, EGFR, and so on [201]. It is worth noting that 5-HT is mostly produced in the gastrointestinal tract by regulating the metabolism of tryptophan hydroxylase-1 via the gut microbiota [202]. It is known that 5-HT plays a crucial role in inflammatory response and immune adjustment as a brain-gut peptide [203]. This may be a reason why BXD exerts multifunctional effects on the brain-gut axis.

## 5. Conclusions and Future Perspectives

Intercellular signaling transduction regulates the life activities of the body. Signals stimulate and then activate signaling pathways that ultimately produce a series of cellular responses, which is a process that gradually amplifies external stimuli. In the organisms of humans and animals, signaling pathways are involved in the occurrence and progression of intestinal inflammation, as are the therapeutic mechanisms of traditional Chinese medicines against intestinal inflammation. Multi-target Chinese herbs and their extracts influence multiple pathogenic factors of intestinal inflammation, including modulating immunity and gut flora and reducing stress brought on by environmental changes. These effects are attributed to the promotion or inhibition of intercellular signaling pathways to decrease the release of associated inflammatory cytokines and suppress abnormal death and differentiation of mucosal cells and immune cells in the intestine. Even Chinese herbal compounds have stronger anti-inflammatory effects through compatibility and contraindication.

For IBD patients, traditional IBD drugs are divided into two groups: anti-inflammatory drugs and immunosuppressants. However, these conventional medications gradually revealed some limitations, including prolonged treatment times and frequent disease flare-ups. Furthermore, biological agents currently approved for the clinical application of IBD, consisting of anti-TNF agents and anti-integrin agents, could overcome the limitations of traditional drugs. However, the economic burden may be a disadvantage to the wide use of biological agents in IBD [6]. Many bioactive components can be found in herbal medications and their extracts, but their identification and pharmacological actions are still mostly unknown. The inability to set uniform standards, the impossibility of measuring the effectiveness of Chinese medicines, and the generally unstable quality control of Chinese medications are the results of this. The effectiveness, patient acceptance, affordability, relative safety, and other advantages of Chinese medicines, however, outweigh their possible drawbacks [204]. In recent years, the Chinese medicine industry has been developing rapidly. Driven by modern science and technology, the multi-pathways and multi-targets of Chinese herbs are more effective and comprehensive in the treatment of IBD under the holistic cyclic view of Chinese medicine.

Nanotechnology can improve drug-specificity, stability, bioavailability, and in vivo distribution, especially for natural compounds targeting colitis lesions. This increases drug concentration at the site of inflammation and reduces side effects associated with systemic absorption of the drug [205]. Due to the synergistic impact created by the usage of the combination of curcumin and chitosan, the in-vivo investigation using the excision wound model demonstrated that curcumin-loaded chitosan nanoparticles speed up the healing process of the wound [206]. The study target in earlier studies on the pharmacodynamic foundation of TCM decoction is frequently macro-integration and is more heavily weighted toward small-molecule bioactive components. The pharmacological effects of decoction are now being intimately linked by researchers to nano-scale particles, such as nanoparticles [207,208]. As an illustration, the particles in the Coptis Chinensis decoction aggregates not only enhanced transcellular BBR transport by active transport and endocytosis but also increased paracellular BBR transport through TJ modulation [209]. Moreover, ginger-derived lipid particles have a synergistic anti-inflammatory impact and can target the delivery of siRNA for the treatment of colitis [210]. Feed additives are dietary supplements that are added to animal feed to help livestock that are unable to obtain enough nutrients from normal meals. Nanotechnologies have the power to supply new vehicles for nutrient supplementation while also enhancing the functionality of feed molecules. Due to their small size, minerals in the form of NPs can enter cells and the intestinal wall more quickly, increasing their bioavailability [211].

For livestock, traditional antibiotic drugs have problems such as drug resistance and residues, which limit their use. While plant extract additives do not easily produce bacterial resistance and drug residues, they are expected to solve the long-term plague through the development of animal husbandry antibiotic residues and food safety. It can provide a reliable guarantee for the feed industry and livestock and poultry farming efficiently. Fed with natural products from medicinal plants, they could improve their production performance, body immunity, anti-stress function, and gut health to encourage growth and boost financial gains. So, natural products with anti-inflammatory effects are quite promising as feed additives instead of antibiotics.

The pathogenesis of intestinal inflammation and the therapeutic mechanism of herbal medicines involve numerous targets and pathways; however, current research focuses mostly on a single pathway, and it is unclear how reported signaling pathways interact with one another. Moreover, the mechanism of action of herbal medicines has not been completely elucidated for the genetic and immune factors involved in IBD. In addition, TCM has complex compositions, and our research should focus on the identification and metabolism of their active ingredients, and the study of their mechanisms of action. This will lead to the rational use of medication and enrich the theoretical basis for the clinical application of TCM and its extracts in humans and the production practice in animals. Overall, natural compounds and formulae play a role in the interaction among cell signaling pathways, immunomodulation, and gut microbiota, and hold promise as supplements for IBD patients and livestock healthcare.

## Figures and Tables

**Figure 1 molecules-28-06811-f001:**
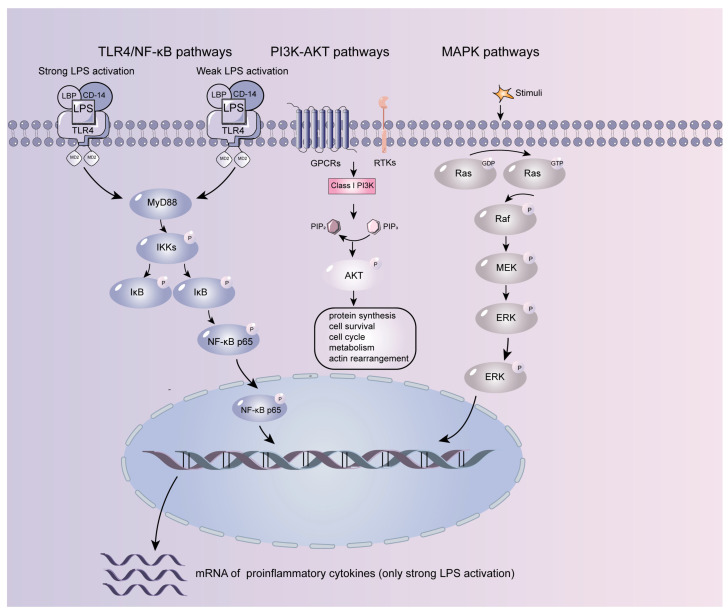
Mechanism of NF-κB/TLR4, PI3K-AKT, and MAPK pathways related to intestinal inflammation [23,24,25,26,27,28].

**Figure 2 molecules-28-06811-f002:**
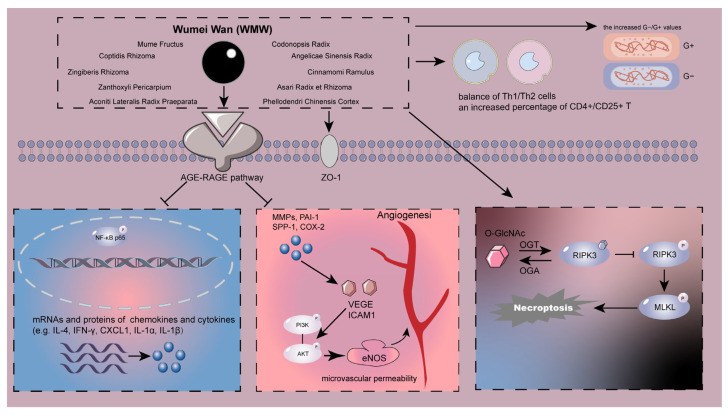
Mechanisms associated with the role of WMW in intestinal inflammation [161,162,164,166,168].

**Figure 3 molecules-28-06811-f003:**
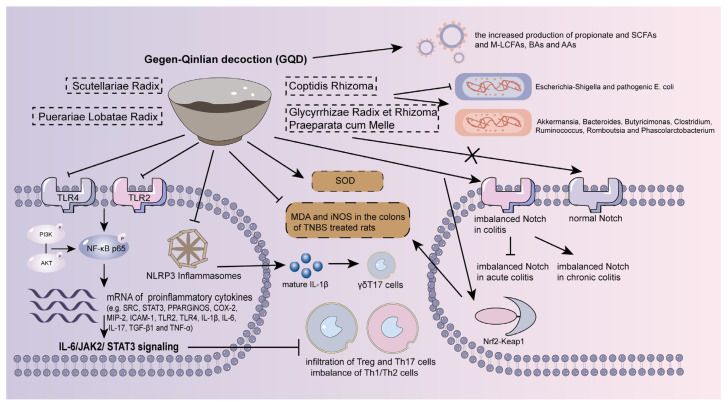
Mechanisms associated with the role of GQD in intestinal inflammation [174,175,176,178,179,182,183,184,185,187,190,191].

**Figure 4 molecules-28-06811-f004:**
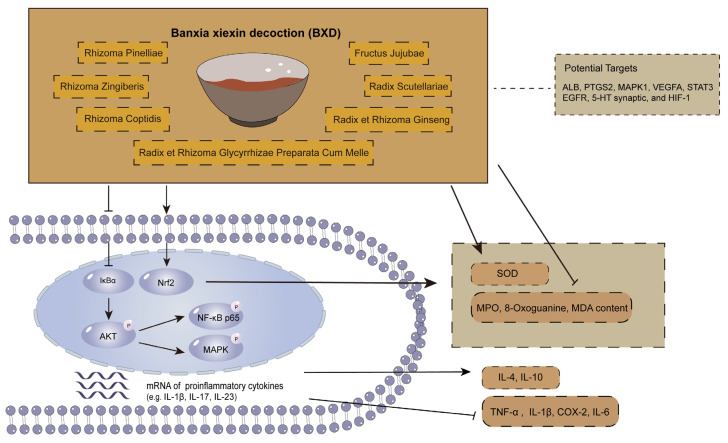
Mechanisms associated with the role of BXD in intestinal inflammation [192,196,197,201].

## Data Availability

Not applicable.

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
