# Peer review of "Protective Application of Chinese Herbal Compounds and Formulae in Intestinal Inflammation in Humans and Animals"

_molecules, 2023, doi:10.3390/molecules28196811_

Round 1
Reviewer 1 Report
In this review titled as “Protective application of the natural product in intestinal inflammation of human and animal”, Yang yang et al., reviewed four cellular signaling pathways in intestinal inflammation (TLR4/NF-κB, PI3K-AKT, MAPKs, and Apoptosis)and summarized the pharmacological mechanism and application of four Chinese herbal compounds (Berberine, Sanguinarine, Astragalus polysaccharide, Curcumin) and three formulae (Wumei Wan, Gegen-Qinlian decoction, Banxia xiexin decoction) against intestinal inflammation.
There are some serious problems in this review:
1, The review lacks of logic. The authors summarized four signaling pathways in intestinal inflammation in Section 2, they supposed to these pathways are very important in the pathogenesis. While in Section 3 and Section 4, when they summarized the pharmacological mechanisms of Chinese herbal compounds and formulae, the mechanisms were not focused on the above pathways at all. So it is difficult to understand why they reviewed the four signaling pathways in this disease. In fact, these four signaling pathways are important physiologically and pathologically in all cells, so to introduce them to a specific disease, intestinal inflammation here, it should be more specific, such as in which cells, and which initial stimuli to activate which targets/receptors to activate the pathways, which are lack in this review and compromise its scientific significance.
2, The review lacks of good design and focal points. The pathogenesis of intestinal inflammation includes diet, microbiota, immune response, breach of intestinal barrier, nerve system in gut, and genetic factors. The review should choose one or two to focus. And the cells involved in this disease include immune cells, nerve cells and structural cells in the gut. The review also should focus on a specific cell type or at least to specify when to introduce the cellular pathways.
3, I could not understand why this review includes four Chinese herbal compounds (Berberine, Sanguinarine, Astragalus polysaccharide, Curcumin) and three formulae (Wumei Wan, Gegen-Qinlian decoction, Banxia xiexin decoction) against intestinal inflammation, but not include cannabionoids, which are from plant cannabis, and its application to IBD has been well studied and their receptors CB1 and CB2 in gut, should be summarized.
4, The review is titled as “Protective application of the natural product in intestinal inflammation of human and animal”. It supposed to include the studied in human and animals. While the pathogenesis of intestinal inflammation in human and other animals are not same, so at least the authors should refer to the difference when quote an animal model which is less common. For example, the treatment of human IBD and dog IBD is different. The former requires immunosuppressive agents and the latter needs to make dietary changes as a common treatment.
And there are many physiological difference in guts between human and chickens.
5, The summary of table 1 is difficult to understand. It is better to group studies in human, in animals and in vitro, or group them as immunomodulation, microbiota, nerve and intestinal barrier.
6, Commonly, “natural products “ do not includes TCM formulae.
I strongly advise the authors rewrite the review and make a focus on specific mechanisms, and not to summarize both compounds and formulae together in a review, which may make it difficult to write and clear to the readers.
Author Response
Dear reviewer:
Thank you for your letter with the comments concerning our manuscript entitled " Protective application of Chinese herbal compounds and formulae in intestinal inflammation of human and animal ". Those comments are all valuable and very helpful for revising and improving our manuscript, as well as the important guiding significance to our researchers. We have studied the comments carefully and have made corrections which we hope meet with approval. Revised portions are marked in yellow in the manuscript. The main responses to the comments are as follows.

Reviewer 2 Report
Please find attached my comments, doubts and suggestions. These comments were made in order to improve the text and its comprehension by the readers.

Author Response

(The authors gave the same response as above.)

Reviewer 3 Report
The present review manuscript is summarizing the scientific knowledge about the mechanisms of action and possible therapeutic applications of several molecules used in Chinese herbal medicine under different formats against intestinal inflammation, concretely berberine, sanguinarine, astragaline and curcumin. The review details many mechanistic studies with the revised molecules and is of interest to the filed. However, there are several important references from 2020 or earlier this year (2023) which also review extensively the topic, e.g. doi: 10.3390/nu15041031, doi: 10.3390/ijms21113956, https://doi.org/10.1016/j.biopha.2022.114086 and these should be included and/or cited in the present work in order to imporve the manuscript and avoid overlapping.
An inclusion of an Abbreviations subsection in the manuscript will help improve the understanding of the manuscript by all readers.
Please, in section 3.1. (line 237) include information about the concrete animal model reviewed.
Please, in the figures designed for this manuscript, include in the corresponding captions information about the bibliographic references on basis of which the concrete figure is drawn.
The conclusions section is summarizing quite well the available knowledge regarding patient care in IBD, however, in several subsections of the manuscript there is only information derived from animal models. Since the title of the manuscript refers to both animals and humans, please, include more information related to human dietary interventions, where available, and if not, please, at the end of each corresponding section include a sentence stating the lack and need for such interventions.
There are numerous grammatical errors throughout the manuscript, including the wrong use of definite article in front of the words, plural and signular forms of the words, verbal forms, etc. For example, there are grammar errors in the title, wrong order in the sentences (line 272, lines 53 - 58, etc.), wrong verb form (line 40, line 121, etc.), wrong use of words (e.g. amount of inflammation istead of level/ intensity of..., economic animals instead of animals of economic importance), a word is missing in line 189 and many more examples existing throughout the text of the manuscript.
Also commas and other punctuations are missing in several lines.
A thorough review by native speaker of the manuscript (not referring to the terms used) will greatly benefit the clarity of this work.
Author Response

(The authors gave the same response as above.)

Round 2
Reviewer 1 Report
I am appreciated that the authors have revised according to specific comments from reviewers, and made the manuscript much enriched. However,it still lasks a logic structure and a focus. It has involved contents such as a certain disease ( intestinal inflammation), some mechanisms( 3 signaling pathways and their relationship with immunity and gut microbiota ), 5 natural products and 3 Chinese herbal formulae .Each section is OK , while there's no clear linkage between sections as a whole. More specifically, please answer these questions:
1, What are the key mechamisms of this disease that are related closely to the 5 natural products and 3 Chinese herbal formulae? And please focus on such mechanisms( from target to signaling , then to function).
2, I still do not know why the authors choose 5 natural products (one was referred in the review report) and 3 Chinese herbal formulae in this review. If this article focused on the TCM, it should summarize generally how many natural products , herbs and formulae used /studied in this disease, clinic or preclinic, and then narrow down to certain promising natural products /herbs/formulae. In this article , the authors listed the results one by one, but summarized not enough, even in the tables.
3, I strongely adverise not to include the 3 Chinese herbal formulae in this article, until its TCM theories can be written clearly. Between the natural compund and formulae, there are many single herbs need to introduce. This article can revise ,focusing on certain mechanisms and natural products, summarized more, not just list the results.
Author Response
Dear reviewer:
Thank you for your letter with the comments concerning our manuscript entitled " Protective application of Chinese herbal compounds and formulae in intestinal inflammation of human and animal ". Those comments are all valuable and very helpful for revising and improving our manuscript, as well as the important guiding significance to our researchers. We have studied the comments carefully and have made corrections which we hope to meet with approval. Revised portions are marked in yellow in the manuscript. The main responses to the comments are as follows.

Reviewer 2 Report
After a thorough analysis of the changes made by the authors, in my opinion the manuscript was accepted for publication in Molecules. I believe that the changes made to the text will enrich the review and the body of knowledge on alternative therapies for the treatment of intestinal inflammation is a chronic gastrointestinal disorder. Thus in the final version of the manuscript they have considerably improved the clarity and soundness of your work. Therefore, I believe that your study will have a valuable impact on the academic community and will undoubtedly stimulate discussions and future research on this topic.
Author Response
Thank you very much for your valuable comments, which make my work better. I wish you all the best in your future endeavors! Good health!
Reviewer 3 Report
The authors have responded adequately to the issues raised previously by this reviewer.
I personally lack as reader some more information in the "Conclusions and Future perspectives" section (from the available knowledge) on the potential of the possible encapsulation in nanoparticles or the development of complex mixtures of the natural products reviewed. This could enhance the effect in IBD patients of the traditional formulations.
Author Response

(The authors gave the same response as above.)
